# Cross-Border Territorial Development through Geographical Indications: Gargano (Italy) and Dibër (Albania)

Antonio Caso [1] and Simona Giordano [2,*]

1    Department of History and Cultures, University of Bologna, 40126 Bologna, Italy
2    Department of Humanistic Research and Innovation, University of Bari Aldo Moro, 70100 Bari, Italy
*    Correspondence: simona.giordano@uniba.it; Tel.: +39-3332622197

**Definition:** In a globalized context, characterized by dominant trends towards the homogenization of food products and taste, local and niche productions play a vital role in creating effective strategies of territorial development. Albanian food heritage is definitely one of the most various of the Western Balkans. The Ottoman domination and the Mediterranean position just in front of Italy led to an incredible mix of cultures and traditions. As Albania is a candidate to join the European Union, it has a stronger opportunity of protecting its excellent-quality food products with PDO and PGI marks. Moreover, Albania's territory shares fundamental features with Gargano lakes, especially with relation to the county of Dibër, where Ulez and Shkopet lakes are located. Both the areas' traditional food products are and can be an important factor of sustainable and participatory development, and the present contribution aims at exploring possible paths of territorial development at a cross-border level, in the framework of a sort of "dialogue" between the two regions through Geographical Indications (GIs).

**Keywords:** cross-border development; Protected Denomination of Origin (PDO); Protected Geographical Indication (PGI); Albania; Gargano; traditional food products

## 1. Introduction

When it comes to Geographical Indications (GIs), the peculiarities that consumers find appealing are connected to the distinctive set of characteristics of each product's specific geographic origin and/or the specific production techniques employed in each site/production area (i.e., the concept of "terroir") [1]. According to Bowen and Muterchbaugh [2], the term "terroir" reflects the relationship between origin and quality by asserting that the quality of each terroir is derived from the choice of particular varieties; although the term originated in the viticultural sector, the extension of grape variety protection to the entire agri-food sector allowed the term to be widely used (Loi n 90-558 du 2 juillet 1990 relative aux appellations d'origine contrôlées des produits agricoles ou alimentaires, bruts ou transformés—Légifrance). According to sociology, this idea conjures up bonds of solidarity that form around common identities, destiny, and skills as well as the formulation of social norms [1].

Therefore, the primary function of GIs in this scenario is to offer a reliable certification method capable of resolving the problem of asymmetric information [3]. In keeping with the overall scope of EU quality policy, the discussion in this contribution, while not claiming to be exhaustive, aims to outline a framework for analysis of the relevant issues related to the preservation of the socio-cultural features of different production sites and traditional know-how, all of which deserve special attention and protection. By adding "Geographical Indication" (GI) to product names, businesses may effectively target specialized markets while simultaneously igniting a positive process that can increase customer confidence and identify high-quality items. Products that are being considered or that have received GI recognition are included in specialized registers along with any pertinent data that may be

used to pinpoint the location and details of each product's manufacture. The GI protection system, which is governed by EU legislation, safeguards the names of a number of products by verifying the presence of distinctive traits and enhancing their reputation as products with a connection to the region of production. PDOs (Protected Designation of Origin) and PGIs are GIs (Protected Geographical Indication). The primary distinctions between PDO and PGI relate to how much of the production process must necessarily take place in the same region or how much of the raw materials for each product must originate from that region [4].

As far as the legal framework is concerned, the main reference is "Regulation (EU) No 1151/2012 of the European Parliament and of the Council of 21 November 2012 on quality schemes for agricultural products and foodstuffs on the protection of Geographical Indications and Designations of Origin for agricultural products and foodstuffs" (Official Journal of the European Union, Regulation (EU) no. 1151/2012), replacing Council Regulation (EC) No 510/2006, that foresaw the possibility for producers from third countries to finalize trade agreements with the European area (Official Journal of the European Union, Regulation (EU) no. 1151/2012).

If the nation of origin of the non-European product name has a bilateral or regional agreement with the EU that provides for the mutual protection of the same product name, then the non-European product name may also be registered as a geographical indication (GI). While both EU and non-EU GIs protected by agreements may be viewed on the GIview site, GIs applied for and formally included in the EU registers can be consulted on eAmbrosia (the official database of EU GI registrations). It is important to keep in mind that although customers are normally prepared to pay more for GI items, there may be variations in the amount of the premium. In actuality, GIs may be considered the cornerstone of an effective differentiation approach. Differentiation is one feasible tactic to establish a long-lasting competitive advantage, according to Porter (1985) [5]. In a differentiation strategy, businesses work to stand out from the competition along certain customer-valued dimensions. As a result of their dominance in this area, they are rewarded with a higher price. The following factors are part of the economic justification for geographical indications: quality uncertainty and asymmetric knowledge may be detrimental to consumers; products with high and low quality may be priced similarly; the "lemon" dilemma, where low characteristics push out high qualities, may occur [6]; local origin influences quality; geographical origin protection might prevent market failure; geographic origin becomes a search characteristic instead of only a credential under legal protection and the related label; regional origin label protection lowers search expenses and improves customer welfare; intellectual property rights; manufacturers of excellent quality are compensated more and earn more money; the market is kept free of copycats and unoriginal manufacturers; helpful for the development of rural areas, rural communities, and the economy.

In the last twenty years, many publications have highlighted the connection between recognized regional food products and economic benefits for local rural areas [7]. In the context of this research, it is critical to analyze how Geographical Indications (GIs) have the ability to encourage agricultural development, which is inextricably linked to the intellectual and cultural property protection that GIs offer to specialist products.

To continue the categorization, GIs can be separated into PDOs and PGIs products based on the various production processes incorporated in each case to be evaluated. According to the EC Regulation 2081/92:

- Protected designation of origin (PDO) is attributed to a product whose characteristics are exclusively linked with a country, a region or a place due to its particular environment features or to its specific cultural traditions. All the phases of production must occur in that specific place.
- Protected Geographical Indication (PGI) is also attributed to a product whose characteristics are exclusively linked with a country, a region or a place due to its particular

environment features or to its specific cultural traditions. Not all the phases of production, however, need to be completed in a specific place, but only one is necessary.

Progressively, the European Union started to understand the possibility of these recognized products for marginal rural areas and the importance of a more diversified agriculture. In accordance with the European Union's stance, three dimensions of rural development are worth considering: endogeneity, integration, and sustainability [8]. It is feasible to increase the economic worth of agricultural products by developing a distinct reputation associated with their characteristics, so capitalizing on a critical marketing tool [9]. Moreover, they can have positive effects also on tourism [10]. The strong cultural identity associated with these products is a potent means of boosting indigenous peoples' and tourists' awareness of the values embedded in them and the natural resources of each product location. Indeed, as Lamy [11] argues, Geographical Indications can operate as a wealth multiplier; they can be regarded as both a resource to be relied on and a part of each community, a kind of "common intellectual property" [12].

GIs have the potential to encourage a more equal distribution of wealth between urban and rural areas by offering rural development opportunities. It is critical, inside the European Union and as a result of the current debate, to stress understanding about the importance of this tool not only in countries where it is already well recognized, such as Italy or France. Generally Mediterranean countries are more involved in the protection of their products, while a positive example for Western Balkans is Slovenia which from 2004 has managed to make 25 PDO and PGI products approved by the European Commission. Even if is not in the top 10, Slovenia is 11th for the number of PDO and PGI wines and any information can be found on the website of the Ministry of Agriculture, Forestry and Food. As the main actors of the next possible enlargement and as the "bridge" for the new commercial routes with Asia, Western Balkans economies have been taken under the observation of European Union and, even though in Western Balkans countries agriculture employment is decreasing, it still plays a fundamental rule: 40.28% in Albania, 19.12% in Bosnia and Herzegovina, 19.02 in Serbia and 7.64% in Montenegro (there is no data available for Kosovo and Former Yugoslav Republic of Macedonia). In addition, Croatia, Slovenia and Greece (respectively 7.54%, 4.93%, and 12.13%) have a higher rate of agriculture employment than the average of EU that amounts to 4.92% (World Bank data, 2018). Therefore, the protection of traditional excellent food products appears to be an important rural development instrument in that area. Some products have been already mentioned in several scientific publications about Serbia [13], FYROM-Northern Macedonia [14], Kosovo and Montenegro. Many products mentioned in the research about possible PDOs and PGIs from Western Balkans are cheeses or fresh and cooked meat products while in FYROM-Northern Macedonia we can find fruits, vegetables, and cereals. The situation of Albanian products is different, probably due to the location of the country between the Slavic countries of the Balkans and the Mediterranean nations like Greece. Moreover, the Albanian territory (which represents 0.26% of European surface) encases more than 30% of European plant heritage [15] and this fact can explain the huge diversity of Albanian typical products. Additionally, the isolation of the country from the Serbian Empire before, and from Yugoslavia in the 20th century makes the traditional food context of the country extremely various. In 2008, harmonized with the EU legislation and with the help of the European Patent Office, the first Law on Geographical Indications of Products and Services has been approved by Albanian Parliament and the General Directory of Patents and Trademarks has been designated as responsible for the procedures of protected origin.

## 2. A Series of Niche Products

In the present section a series of niche products are explored [16,17], as they hold a strong potential in view of effective strategies of local development with reference to Albania, through an ad hoc recognition of GIs mechanisms. The following map shows the location of a series of products described hereunder (Figure 1)

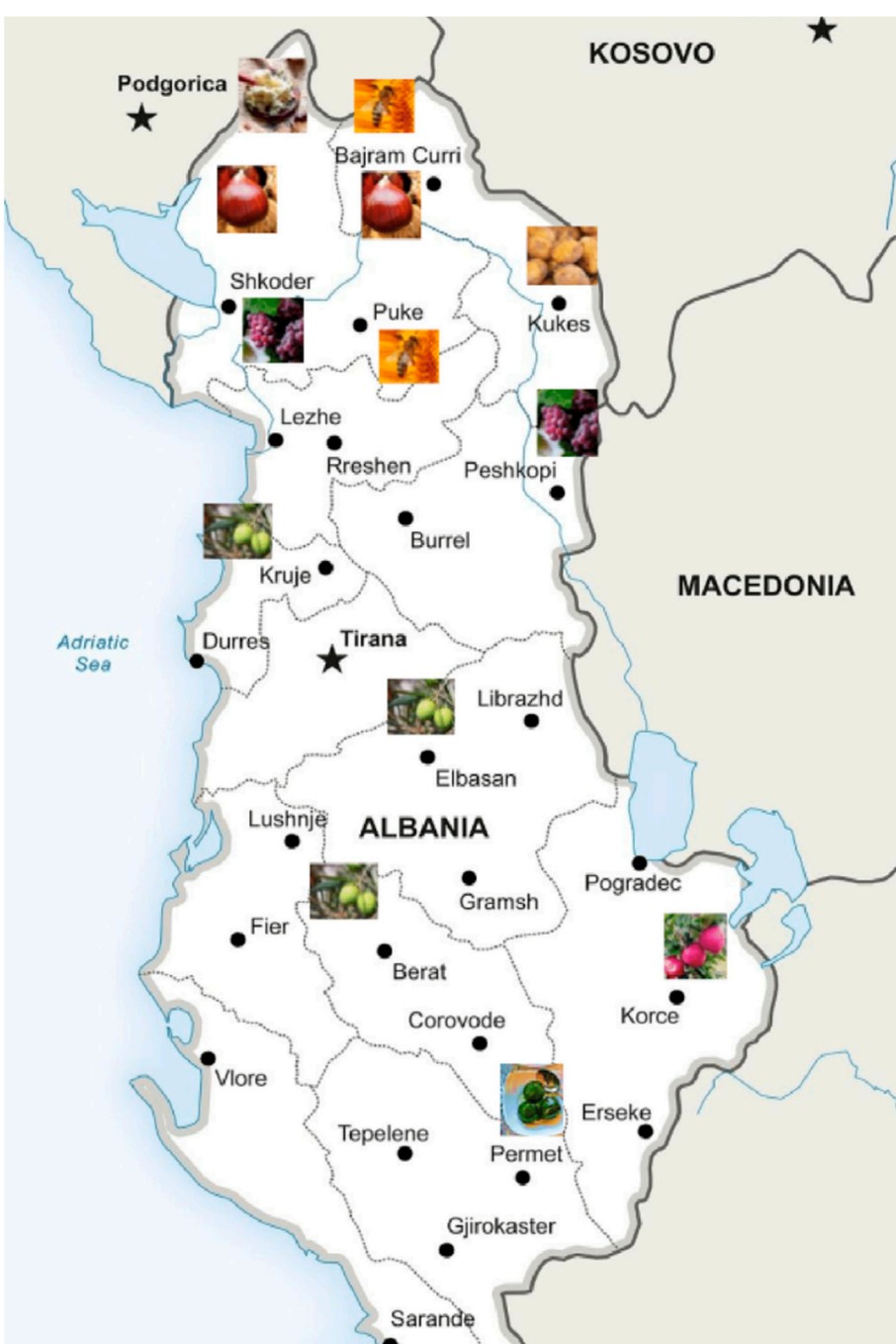

**Figure 1.** Authors elaboration; map of main Albanian GIs.

*2.1. Northern Albania*

The traditional products of Northern Albania are linked with the natural area of the Valbona National Park and with the mountains at the borders with Kosovo and Montenegro. The products are typical of mountain areas like chestnuts and honey, similarly to the northern countries of the Western Balkans. Mishavin, instead, is made by a catholic Albanian tribe which from centuries has inhabited the area of Kelmendi at the border with Montenegro. The wine "Kallmet", instead, comes probably from the homonym village in Lezhë County. It is cultivated above all in Dibër County at the border with Northern Macedonia and in the area between the Adriatic Sea and the Shkodër's Lake.

### 2.1.1. Chestnuts from Reç and Tropojë

Chestnuts are widespread all around the Balkans. In Albania, chestnut trees are located in the northern part of the country, in Tirana's and Pogradec's districts chestnut trees covering a total area of approximately 14,620 ha [16]. Chestnuts from Northern Albania are the only ones prepared to receive the protected designation in accordance with the regulation of 2008 and two possible PGI chestnuts are found in Shkodër and Kukës counties, in the areas of Reç and Tropojë. The chestnuts of Rec have been renowned for about three centuries as an indigenous product of this area of Albania. The chestnut grove includes Reç and the villages of Qafë-klasë, Mucaj, and Doç and is located 280–600 m above sea level. The surface of the chestnut forest amounts to around 600 ha with 15,000 chestnut trees. The production consists of 500–600 ton/year of which 95% is exported to Italy. The fruit contains 15–16% sugar [17]. In 2013, farmers asked for a denomination of origin and, therefore, only traditional methods are permitted. The difficulties are linked to the lack of adequate communication routes. In the area of the chestnut grove, in fact, only 30% of the streets are paved [17]. Nevertheless, the first cooperative of producers, called "Reçi Prodhimtar" has been already created. Chestnuts from Tropojë, instead, are harvested in the forests around the city where one of the largest chestnut groves in the Balkans is located, a few kilometers away from the Valbona National Park, which includes seven municipalities and about 50 villages in about 2400 hectares. A total of 72% of the population lives in rural areas. The chestnut forest consists of 2500 ha and around 225,000 trees [17]. The principal massive are located in the higher part of Bajram Curri and the massive of Kernaja [18]. Thanks to the discovery of some archaeological relics from the Illyrian age, it is possible to affirm that the collection of chestnuts is, in this area, a millenary practice. These chestnuts are 2–3 cm in size and they are renowned for being particularly sweet. Due to the short distance, many farmers export their product in Kosovo, while professional traders usually sell it in Tirana and Shkodër. Another important partner for this product is Italy since Tropojë chestnuts are the only Albanian chestnuts which fulfill the quality standards required for the European Union [19].

### 2.1.2. Chestnut Honey and Honey from Pukë

Within the products most popular in Albania, honey is one of the most consumed. An important factor to consider is the attitude of the Albanian people towards honey and the bees themselves. It is enough to say that in Albanian there are two terms to indicate death: one for animals (zof-ngordh) and another one valid for human beings and bees (vdes).

Annual per capita honey consumption amounts at about one kilo, twice as much compared to Italy, with a selling price of around 8 euros/kg, at the time of production. Starting from the 1990s, production faced a transition period and experienced an increase of 800% [20]; the north is the protagonist in production, with particular regard to the areas of Zadrima, Tropojë, Reç, and Pukë. The main features of the cited areas relate to the lack of infrastructures and connections, especially if compared to areas located in the Centre and South of Albania.

It is of particular interest to consider as a potential candidate for the recognition of a GI is chestnut honey from Tropojë and Reç and, in the same forests of Tropojë, the so-called "chocolate honey", a variety of chestnut honey distinguished by a color far darker and a taste predominantly sweet, whose production season is June [17].

Moreover, honey produced in Pukë embeds a peculiar character derived from the various medicinal herbs located in the same area; as described above, production areas feature a series of weaknesses such as the small size of businesses, poor communication infrastructures and innovation costs necessary to render the whole sector more competitive [21]. Despite the cited elements, production is increasing, and sales occur mainly in the markets of Tirana or Durrës. Born in 2001, the association AgroPuka aims at enhancing the mentioned production activities and includes around 60 farmers from the Pukë region. The association accompanies producers in the promotion of traditional products through the participation in national fairs and specific events [22].

### 2.1.3. Potatoes from Kukës

Potato cultivation is quite common in several places of Albania. Mostly in Korçë area, Elbasan [23], Bicaj, and Kukës. Specifically, the last one is renowned for its quality. This local variety (also called "White Potato") is characterized by its white color and its particularly sweet taste. Usually, due to the nearness of the border, they are sold in Kosovo markets. The yield of potatoes in Albania is quite good compared with some other countries. It amounts to 245,894 hg/ha, more than Serbia (178,120 hg/ha) and Greece (237,800 hg/ha) even though they have a higher production [24]. The interest in agriculture is rising in recent years; as a result, the Kukës County sector enterprises raised from 22 in 2012 to 399 in 2016 with a growth of 1713.6% [23]. Nevertheless, one of the main problems is the fragmentation of farms which usually puts an obstacle to the creation of cooperatives. Other considerations need to be made about the low level of mechanization, such as in Kukës only five agriculture tractors have been registered in 2016 [23], and the irrigation capacity, which according to the data, only the 7% of soil can be irrigated from the reservoirs [25]. Another obstacle is given by the high cost of transport; the Kukës area is the 4th highest of Albania for agricultural products and there is also one point of collection with refrigerating room near Kukës [25]. Other collection points and the adoption of current technologies could be a good starting point for the promotion of this product.

### 2.1.4. Mishavinë

Mishavinë is part of the "big cheese" family, a variety spread along the entire Balkan Peninsula and in the Anatolian region. It is a grated cheese, left to ferment in a barrel with cleared butter called tylëne [26]. In Albania, it is produced in the Kelmend region, in the extreme north of the country, which stands as one of the least known regions in Europe. The strong isolation and the lack of jobs have led to a high level of emigration, but also to the possibility of keeping many traditional productions and uses alive. Mishavinë is produced exclusively in the summer months and is consumed during the winter. The curd is obtained from a mix of sheep and goat's milk which are raised in the pastures of Mount Trojan. The color of the cheese varies from white to straw yellow and its structure gradually becomes denser through aging. It is characterized by a buttery consistency, its flavor is reminiscent of the herbs growing in the mountains of the Kelmend region, and that becomes spicy over the months. This product is already a Slow Food Presidium. The Presidium is the direct consequence of the initiatives of the NGO VIS Albania, and it is supported by the Slow Food Chefs' Alliance. It was established in 2015 as part of the activities of ESSEDRA, a project co-funded by the European Union and by Slow Food with the purpose of strengthening the integration of civil society with Europe in the Balkans and Turkey. The specific goal of the Mishavinë is to establish a protocol of actions in order to let this product meet European food-safety regulations. The production area, however, is difficult and the lack of adequate communication routes is certainly one of the major obstacles. On the other hand, the product is unique, and the area is very close to the border with Montenegro and not far from the one with Kosovo and this could be very convenient for export purposes. It is almost unknown in the other regions of Albania, and, with an appropriate marketing campaign, it can be a solid brand in a country where the "white cheese" and the so-called "kaskaval" are almost the only ones on the market.

### 2.2. Southern Albania

The traditional products of Southern Albania include the Kokerrmadhi from Berat (which has been described in the previous paragraph), apples from Korçë and a dessert called gliko. This area is the most related to the Greek tradition as it is possible to notice from the last product. Many citrus fruits, olives, and grapes are grown along the Adriatic coastline mostly in the area of Sarandë, but there is not the necessary evidence to establish their peculiarities in order to become a recognized product. Other traditional products are the black goat from Dukat and the raki from Skrapar, but more research is needed to include them in possible PDOs and PGIs.

### 2.2.1. Apple from Korçë

This variety of apple is particularly renowned in Albania for its sweetness. Korçë County is one of the largest producers of apples and this sector is growing rapidly. The soil occupied by apple trees is 1850 ha with an annual potential of 40,000 tons representing 65% of the apple production of Albania [27]. Additionally, Korçë has experienced a huge increase of agricultural enterprises over the last decade. In 2012, there were 190 companies, while in 2016 they had become 6242 [23]. A tree can produce up to 200–300 kg of fruit and apples can be stored for over six months after the harvest in the centers for the collection of apples spread around the county. The life expectancy of a grafted tree is about 60–80 years. This high-quality product needs proper branding and protection from falsification which, given the product's fame, is particularly widespread on the national market. Currently, there are 11 associations of farmers [27]. Another opportunity is given by the rising tourism in the area. Every year, there is a festival celebrating the beginning of the harvest during which farmers show different varieties of apples. The "Applause Party" takes place in the village of Dvoran and tries to attract as many tourists as possible.

### 2.2.2. Gliko from Përmet

Gliko is essentially a technique used to conserve vegetables through sugar. This technique perfectly preserves the shape, consistency, and composition of the fruit and it is used in all the Balkans as well as in some areas of the Middle East and the Russian regions bordering the Black Sea. It can be made with grapes, berries, apricots, mandarins, figs, watermelon, but also with vegetables such as cherry tomatoes and dried fruit such as pistachios. It is usually served as a dessert and a particular variety can be found in Përmet, a city in southern Albania. Here, the gliko is prepared with cherries, aubergines and wild figs, but the most traditional and common is the one made with whole green walnuts. The fruit is left to dry for an hour and is then placed in cold water with lemon juice. Then, sugar is added and the water boils for one hour, while lemon juice is added gradually. Once the fruit has absorbed the syrup, the mixture is filtered into glass containers. The production of gliko was helped by Slow Food (currently Presidium) and the Italian NGO CESVI (Cooperation and Development). All this made the creation of the "Pro Përmet" Consortium and the "Enhancement of the Tourist Environment and its typical products" project possible. The production of gliko can become a fundamental resource for the development of the local economy, but it needs a stronger branding and adequate certification. The product appears to be perfect for export and it is highly eco-friendly since, in this way, it is also possible to use fruit that would otherwise be wasted. The following Table outlines data regarding crops production in Aòbania with regard to different products above described (Table 1).

**Table 1.** Albanian crops production quantity (selected products).

| Product | 2019 (Tons) | 2020 (Tons) |
|---|---|---|
| Apples | 105,933 | 102,167 |
| Chestnuts (in shell) | 5846 | 5616 |
| Grapes | 189,904 | 199,069 |
| Olives | 98,313 | 131,971 |
| Potatoes | 260,661 | 254,886 |
| Oranges, lemons and limes | 15,631 | 17,026 |

Elaboration from FAOSTAT at FAOSTAT.

## 3. The Typical Products of Gargano and the Potential of Dibër for Development Opportunities

In the framework of the proposed dialogue between the two regions at stake, the present paragraph aims at analyzing the typical products of Gargano, that could become a sort of compass for Albania. These products, being already protected by a GI, embed a

series of characteristics and represent a driving force for development. They can become "twinned products" and guide sound policies at a regional level in Albania.

### 3.1. Gargano GIs and Traditional Food Products

### 3.1.1. Orange fruits from Gargano PGI

The laurel trees and century-old holm oaks that line the Gargano shore between Vico, Rhodes, and Ischitella provide protection for the Gargano blonde orange plant. The orange groves are referred to as "gardens" and appear to be as such while orange trees flourish along the slopes. Low barriers separate the gardens, resulting in terraces where oranges are planted downstream, and lemons are cultivated further up. This has been performed in Capitanata for centuries. In reality, it appears that the Saracens originally introduced the melangolo, the first citrus fruit to reach Europe, in this region in the seventh century AD 50 [17]. The first recorded records were made in 1003 when Melo, prince of Bari, brought "pomi citrini" that grew on the Gargano to distant Normandy. Mostly bitter oranges were provided as evidence of the region's exceptional fecundity. Vico del Gargano and Rodi Garganico both rose to prominence in the 17th century as leaders of brisk trade down the Adriatic with Venice [28] and particularly with America. Citrus fruits were really transported by train from Manfredonia to Naples and then transported by sea to the American continent. Despite having just approximately 1000 acres of land, trade flourished during the 19th century to the point that the Gargano region ranked first for profits and third for citrus output in Italy. The oranges received the PGI trademark and Slow Food presidium in 2001, owing to the Consortium for the Protection of "Gargano Agrumi" and the efforts of the Gargano National Park Authority.

### 3.1.2. Lemons from Gargano PGI

A unique kind may be found in Puglia along the Gargano coast. It is known as the Femminello lime because of its high production. grown by Ischitella, Vico, and especially Rodi Garganico. It can produce up to 4.5 floral arrangements every year and is one of Italy's oldest varieties. The Apulian monk writer Filippo Bernardi wrote at the end of the eighteenth century that "the Venetians and the Schiavone come to care for wine, oranges, lemons . . . ; at Rodi one can say that there is a plain of gardens for the quality of the oranges and lemons that there are planes so wide that appear to be either oaks that agrumi". Vincenzo Ricchioni also described the agrumi of the Gargano in his 1811 work "*Statistica del Reame di Napoli*", citing a map from 1808 that provided the value of the entire combined aggregate in about 10.000 "ducati", and adding that if they were exported for 60.000 "ducati" to the outside, for 20.000 to the king, and for another 20.000 to be consumed domestically in the country [29] (p. 73). The Femminello lemon is then divided into two varieties: the "lustrino", or "gentle scorza", and the "fusillo", which has an oblong shape. The first kind is lighter and has a more delicate peel, but both are distinguished by their high levels of juice and the tenacity of their peel. This blend has been a trademark of IGP since 2007 in addition to being a Slow Food72 product. The coastline of the Gargano "giardini" is covered by limoneti, and the huts are distinguished by their "squarched" chimneys [17].

### 3.1.3. Eel from Lesina

In Apulia, the Mar Piccolo basin of Taranto, the Lakes Alimini and the Brindisi coast are prime spots for these animals, but the absolute richest waters are those of the Lesina and Varano lakes, on the promontory of the Gargano. In the waters of these lakes, eels arrive through the Acquarotta canal and Schiapparo estuary. They are fished here only when they reach commercial size (between 20 and 25 cm). Traditionally, fishing was practiced in pagghiari, thatched huts the scene of autumn fishing while waiting for the traps of nets, the so-called "paranze," to do their work. It seems that disputes related to eel fishing also arose throughout history, especially among monastic communities. One dispute would have occurred between the communities of Montecassino and the Tremiti and another would possibly have arisen in Brindisi between the canals then called Delta and Luciana (currently

known as Fiume grande and Fiume piccolo) between the monks of Sant'Andrea in Insulam and the episcopal curia of the capital. The passion for eels in this area was shared also by the "Stupor Mundi" [17] (p. 100). According to a local story, Emperor Frederick II once wrote to the curia of Foggia to explicitly request a preparation of eels alla askipeciam (now known as scapece; the fish is marinated in vinegar) to Berardo, his personal cook.

### 3.1.4. Farrata from Manfredonia

Farrata is a savory flatbread typical of Manfredonia and its ancient hamlet of Siponto. The use of spelt is attested a great deal in antiquity, especially by the Romans, who used it for flatbreads and baked goods. In this case, we are talking about a round-shaped rustic prepared with spelt, sheep's ricotta cheese, marjoram, cinnamon, salt and pepper, and baked after being brushed with the yolk of an egg. Traditionally, it was prepared for the Carnival of Manfredonia (also called carnevale dauno or carnevale sipontino), but today it can be found at other times as well [17] (p. 94).

### 3.2. Ulez and Shkopet Traditional Food Products

In the framework of the present contribution, it is of particular interest, as mentioned above, to propose an intertwining of the two regions of Gargano and Dibër. These two territories show numerous similarities, as they both possess natural reserves gifted with precious resources derived from the mountains, forests and lakes present (Lesina and Varano on one side, and Ulez and Shkopet on the other one).

The possibility to create an effective dialogue, as proposed in this paper, in order for Albania to take a lesson from Italian GIs, represents a fundamental opportunity towards sustainable and participated paths of territorial development.

In particular, the following products are worth fostering, as a lever for the cited development. As a matter of fact, Albanian produce represents the precious outcome of artisans and family businesses that constitutes the backbone of the national agricultural sector.

### 3.2.1. "Kallmet" Wine

Wine in Albania is characterized by indigenous varieties and a very old tradition. Currently, it is possible to divide Albania into 4 producing regions:

- The coastal plains that include Tirana, Durrës, Shkodër, Lezhë, Vlorë, Fier, Lushnjë and Delvinë. Many vineyards in the area are planted at an altitude of 300 m;
- The central region which includes Krujë, Gramsh, Berat, Përmet, Elbasan, Mirëdita and Librazhd. The altitude of the vineyards varies from 300 to 600 m;
- The valleys and areas close to the eastern mountains that include Pogradec, Peshjopi, Leskovic, and Korçë. Here, the vineyards are found starting from 800 m;
- The mountainous regions where there are no large urban centers, and the terrain is generally clayey and characterized by various depths. The grapes are grown starting from 1000 m.

The main grape varieties include: White (Shesh i Bardhë, Debin e Bardhë, Pulës) and Red (Shesh i Zi, Kallmet, Vlosh, Serinë, Debin e Zi). The most representative is definitely the Kallmet. It is one of the oldest autochthonous grape cultivars occupying about 9% of the national vineyards [30]. It is cultivated mostly in the north and north-west Albania. It is planned to cover at least 20% of all the red wine cultivars vineyards [30]. Other names are "Kadarka", "Scadarka" and "Nero di Scutari". It is the noblest strain for black grapes in Albania and takes its name from a subdivision of the municipality of Lezhë in the north-west of Albania. This ancient variety has violet berries, and its productivity is usually in a 2/1 ratio for a 13–15° alcohol content wine. The grape must contain 67 mL/100 g fresh grape with 21% sugar [30]. Since the 1990s, 64 producers in northern Albania have joined a consortium. The recent international interest for Kallmet as well as for its production has increased so much that, in the Albanian pavilion of the Bio-Mediterraneum Cluster at the Milan Expo 2015, there was an event called "Kallmet: a grape, a wine, a territory". The event was organized by the Ministry of Agriculture and Rural Development of Albania, by

the NGOs Celim Milano and Livia and by the Agency for Investment and Development of Albania. During the event, it was possible to illustrate the initiatives taken in favor of the creation of the protected origin mark, the selection of the Albanian cultivars and the objectives set by this sector of the national economy.

### 3.2.2. Kokerrmadhi Olive

The millenary presence of olives in Albania has been shown by the records of olive export to France dating to 300–150 BC [31] and by the archeological remains by 6th century AD [32]. The genetic patrimony of the country is considered very rich in this sector [33,34] and today 22 olive cultivars are used in the olive plantations of Albania [35]. Kokerrmadhi, together with Kalinjoti, Kallmet and other minor varieties are considered autochthon of Albania [36] also due to the independent clustering of the majority of them from Greek, Italian and Turkish cultivars [37]. Olives and oil have always been an important sector for the Albanian economy. In 2008, the government launched a campaign with the purpose of planting up to 25 million olive trees over several years [38]. Currently, the average production is estimated at around 11.2 kg per tree [39]. This is a relatively low production mainly due to the lack of adequate technology. The main varieties are "Kalinjot" or the "Kokerrmadhi" of Berat, Elbasan, and Krujë. Together, these varieties represent around 60% of table olive production. Ninety percent of the production is concentrated in the districts of Tirana, Valona, Elbasan, Fier, and Berat. According to recent assessments, it is possible to reach 25 kg per tree with adequate irrigation and suitable treatments. Furthermore, the tradition of not using industrial pesticides and fertilizers has created various opportunities for eco-friendly and organic products. The Kokerrmadhi variety is particularly interesting. It is a production that dates back to the Middle Ages and is considered a variety that can be used both as table olives and for the production of oil. The olives are round and medium in size. The trees are mostly located in the hills north of Elbasan and in the villages of Kugan, Shrigjan and Bathes as well as the version of Berat and Lushnjë (which will be mentioned in the part about Southern Albania) and Kripsi i Krujës. In Elbasan and, above all, in Berat this sector took part in the incredible growth of the agricultural sector of Albania in the last decade. From 2012, in Elbasan the number of agricultural enterprises increased from 87 to 3951 in 2016 with an increase of 4441.4% [23]. This growth is still more visible in Berat where in 2012 there were 60 agricultural enterprises while in 2016 their number had increased by 5658.3% and 3455 enterprises operated in that sector [23]. The oil is used in traditional dishes and salads and, currently, the production is about 90–100 tons of olive oil per year for restaurants, markets, and domestic consumption.

### 3.2.3. Jufka

Pasta cooked by hand and traditionally prepared is called jufka. Typically, it is prepared with premium whole durum wheat (Triticum durum) flour that has been processed in water-powered mills. In a bowl, combine the flour, milk, eggs, and salt. The dough is allowed to organically ferment for 5 days in a dry location. The dough is subsequently flattened out, sliced into thin, ribbon-shaped noodles, and allowed to dry. The jufka can be kept in paper boxes, bags, or wooden boxes after drying. To avoid mold, it is critical that the noodles have enough ventilation. One of the distinctive foods of the Dibra region of eastern Albania, jufka is prized for its flavor and handcrafted preparation. Jufka has also been found among the Arbresh population, a group of Albanians who have been residing in Italy since 1480; this places jufka as a traditional, centuries-old Albanian good. In Dibra families, women often cook it. A number of women from the hamlet of Vakuf have taken the initiative to create an artisanal production of jufka for commercial sale due to the lack of job prospects in the region and the desire to introduce jufka to a larger audience.

## 4. Discussion and Conclusions

Geographical Indications, as marks of designation for food products, can be used in different ways [40–43]. The current standards of production of the European Union are

not easy to reach, however, this might be viewed as a significant challenge to enhance the overall Albanian agricultural sector [44]. As a further example, the spearhead of Albanian olives is surely Kokerrmadhi from Krujë thank to its resistance and its productivity. This is the best variety for export due to its characteristics. Kokkermadhi from Berat can be exported too, but it can be also used in order to develop a sustainable tourist sector focused on the best local food.

The situation is certainly diverse. Northern research and the establishment of PDO and PGI marks are more advanced, owing primarily to local NGOs and organizations that have already established a form of Disciplinary of Production for various items such as as chestnuts and honey. The key issue is a lack of appropriate technologies and a lack of good local roadways for transportation. Moreover, the nearest harbor (in Durrës) is not easy to reach from many villages where the production is held. Nevertheless, these counties and districts are closer to the borders with Montenegro and Kosovo, which is an advantage in many circumstances. In Central and Southern Albania, the techniques are usually better, but there are less possible recognized products. Another important economic source linked to these products is tourism. According to the current data, Food & Drink Services sector is the most valuable in Albanian tourism and it contributed 25,515 million lek in 2016 (in 2013 it was 19,600 million lek) [40]. Additionally, the number of employees of Food & Drink Services tourist sector switched from 10,572 in 2012 to 23,732 in 2016, equal to 79% of the total employees of the national tourist sector [40]. Finally, it is important to underline that the potential socio-economic impact of recognized food products needs adequate considerations also of a wider environmental policy [45,46]. PDOs and PGIs have to be seen not only as intellectual property rights but as part of a broader policy which takes into account local identities and natural heritage which can assume a fundamental rule from a social, economic and environmental point of view [47].

It is of particular interest to conduct an initial SWOT analysis (Table 2), in order to investigate the possible impact of the described development opportunities through the introduction of GIs and the enhancement of traditional food products, as follows:

**Table 2.** SWOT analysis.

| Strengths | Weaknesses |
|---|---|
| The development model realized from the enhancement of traditional agri-food products and GIs can certainly provide particularly interesting implications from the point of view of territorial development. Especially the area on the border between three different districts (Durres, Lezha and Diber) would stand as a center for the valorization of agri-food products from all three districts. | The development model realized from the enhancement of traditional agri-food products and GIs can certainly provide particularly interesting implications from the point of view of territorial development. Especially, the area on the border between three different districts (Durres, Lezha and Diber) would stand as a center for the valorization of agri-food products from all three districts. |
| Threats | Opportunities |
| The main threat lies in environmental protection versus product enhancement through food and wine tourism. Especially, the waters of the lake are already subject to rather intensive fishing, which would get worse. In addition, the area is rich in forests that should be protected in any way with respect to tourism initiatives. | The main opportunity lies, without a doubt, in the possibility of enhancing the traditional agri-food products of the area through a quality and sustainable tourist offer, making the Ulez and Shkopet Lakes a true "embassy" of the products of the three aforementioned districts. |

Authors elaboration.

The described cross-border path of territorial development, according to the importance and the interest of examined contexts, opens the way to further reflection to be conducted in the framework of future and cross-border cooperation projects [41,48]. As products included in the present contribution, with their potential to foster rural develop-

ment, represent the outcome of an agricultural sector mainly based on family businesses, it is of particular interest to conduct in-depth research related to the different issues concerning succession in the same family businesses. As described by Więcek-Janka et al. [42], the succession process embeds a series of vital aspects, also connected with emotions associated with it and collective and individual goals that owners determine in view of foreseen changes. In the perspective of the present analysis, the cited analysis proves to be of strong interest in relation to the Albanian context, mainly based on micro-small family businesses. As a matter of fact, Albania, in the aftermath of the initiation of a difficult transition to a more open-market economy, is facing cogent challenges such as rapid urbanization, overuse of land and of natural resources. It is vital to engage in a deep reflection regarding new possible models of governance capable of coping with issues related to the cited scarcity of natural resources and the threats connected with population growth, changes in consumption patterns, climate change and land degradation. In line with data from the World Bank [43], in Albania, agriculture accounts for 19% of the national GDP (2020); this figure witnesses the extent to which the sector at stake proves to be crucial for the whole country, with particular reference to Micro, Small and Medium sized Enterprises (MSMEs), and to which it is fundamental to implement sound and effective people-centered land governance, as several rural dwellers face food insecurity and poverty, being left with limited access to land itself [44,49].

The mentioned discourse, in which GIs play a crucial role, is vital as well in the perspective of fighting phenomena such as land grabbing, mainly deriving from international actors such as China. Finally, it could be of particular interest to conduct further research in the sector at stake by means of tools arising from different disciplines, such as mathematics, in order to broaden the adopted perspective [50–52].

**Author Contributions:** Conceptualization, A.C. and S.G.; methodology, A.C.; validation, A.C. and S.G.; formal analysis, investigation and resources, A.C. and S.G.; data curation, A.C.; writing—original draft preparation, A.C.; writing—review and editing, S.G.; supervision, A.C. and S.G. All authors have read and agreed to the published version of the manuscript.

**Funding:** This research received no external funding.

**Institutional Review Board Statement:** No statement to provide.

**Informed Consent Statement:** Not applicable.

**Data Availability Statement:** Data are authors research's outcomes.

**Conflicts of Interest:** The authors declare no conflict of interest.

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
