# Peer review of "Cross-Border Territorial Development through Geographical Indications: Gargano (Italy) and Dibër (Albania)"

_encyclopedia, doi:10.3390/encyclopedia2040127_

Round 1

Reviewer 1 Report

The paper is valuable and views present and historical sides of a lovely part of the Balkans.

I recommend the authors

--- to add some more quantitative elements (tables, diagrams, formulas),

--- let 3 experts on quantitative and formal sides check the paper and give you a hand,

--- the same with content sides, in particular (need not be too long):

--- consider threats in the regions (dominations of various kinds, such as Turkish daily soaps, etc., Chinese influence in the region (buying up ...), etc.

--- the unique potential to contribute to Europe (e.g., by the vats experiences of its nations),

--- problems with migration streams, etc.

--- role of small and medium-sized companies (and support for them).

See works by Ewa Wiecek-Janka, Berat Kjamili, S. Zeynep Alparslan Gok, Joanna Majchrzak, Gerhard-Wilhelm Weber.

E. Wiecek-Janka, J. Majchrzak, M. Wyrwicka and G.-W. Weber, Application of Grey Clusters in the Development of a Synthetic Model of the Goals of Polish Family Enterprises' Successors, Grey Systems: Theory and Application (May 2020); DOI 10.1108/GS-12-2019-0062; https://www.emerald.com/insight/2043-9377.htm.

Author Response

Dear Reviewer,

As authors, we do thank you for your valuable work and for the suggestions to improve our paper. We feel honoured to follow your remarks, as hereunder outlined.

- A table regarding the production in Albania, of the products included in the paper.

- An expert in the Department has been contacted to check the paper, and we received a positive feedback.

- We inserted some considerations regarding Chinese influence in Albania and role of MSMEs in the country, as suggested.

- We included mention to the work by Ewa Wiecek-Janka et al., with relevant analysis on succession issues.

Reviewer 2 Report

This manuscript gives an overview of Geographical Indications (GI) in Albania and outlines their potential in rural development. The paper uses the region of Gargano (Italy) as a reference point and road map to follow by Albania to promote GI produces. The study is conducted on a spatial basis, and it is structured regionally, using GI as units of study. The paper is carefully crafted, and shows comprehensive information of the area and topic analyzed.

The manuscript is a good input for food, rural development and geography studies, and a positive asset for Encyclopedia.

1. Introduction

This section introduces the reader to the main topic of the manuscript pleasantly, and provides a general framework for the paper. The concept of “terroir” is pertinent in this framework, and the authors have been able to articulate it properly and to link it to the GI content smoothly.

The last paragraph of this section introduces the geographical setting of Albania and the Balkans. Since geography is a key component in GI approach, I would suggest creating a section on its own (e.g. Albania: geographical context) to develop the spatial settings and flesh out geographic traits and singularities. This new section could use the contents in the last paragraph of section 1 and develop some key points. It would be appropriate to complement this new section with a map(s) showing Albania in its context, and show with fluxes or diagrams the spatial influences.

2. A Series of Niche Products

This section gathers and develops GI produces in Albania. It would be illustrative to have a map to know the location of the GI indicated. The authors might also find useful to create a map composition showing photographs or draws of the GI produces described.

3. The typical products of Gargano and potential of Dibër

This section focuses on Gargano (Italy) products to provide “a sort of compass for Albania”. This section, interesting though it is, seems somehow contrived within the manuscript. To avoid the reader distracting their attention and to focus on a clear line of research, I would suggest the authors to modify this section. The new section should use the same contents existing in the current one but modifying their approach so it is more evident that the focus is in Albania (and what Albania can learn from Gargano). I would suggest:

a) Changing the title of this section to emphasise development opportunities

b) Briefly reflect on rural development (perhaps retrieving key ideas already presented in section 1 –line 100-)

c) Expose Gargano produces as example (perhaps just synthesising the current section)

d) Reflect on Albanian produces (perhaps just slightly modifying section “3.2. Ulez and Shkopet traditional food products”)

4. Discussion and Conclusions

This section synthesises the main contents of the manuscript and provides final judgments. Correct though it is, I would suggest adding a matrix to show the main points of the SWOT analysis graphically. I would also modify the contents of the last paragraph (line 523) in order to focus emphasis on rural development in Albania by means of GI, rather than on cross-border cooperation.

0. Linguistic remarks

I am not qualified to judge on the English language used in the manuscript. However, I sense that some sections lack of academic style or adequate language (e.g. line 69 (“in actuality”, 116 (“in fact, Slovenia”), 118 (“any”), 193 (“kg3”), 253 (“the”), 423 (“(nowadays) France”)).

I would recommend the authors to use MDPI’s Language Service for a final English check. As author, I used MDPI’s Language Service in previous occasions and I am pretty satisfied with it as its standard English review includes checks on spelling, style, and in-house guidelines.

00. Title

I would suggest the authors changing the current title for another one focusing on potentialities of rural development in Albania by means of GI, instead of on cross-border development.

Author Response

Dear Reviewer,

As authors, we do thank you for your valuable work and for the suggestions to improve our paper. We feel honoured to follow your remarks, as hereunder outlined.

1 and 2. Introduction and A Series of Niche Products

We added a map with the location of the GIs indicated, and we thank for the precious suggestion. As to the spacial influences, we do not feel like including them as we do not feel enough expert with regard to this aspect.

3. The typical products of Gargano and potential of Dibër

As you suggested

- we changed the title of the section to emphasise development opportunities

  • we inserted a brief reflection regarding Albanian produces (perhaps just slightly modifying section “
  • 3.2. Ulez and Shkopet traditional food products”)

4. Discussion and Conclusions 

As suggested, we put the SWOT analysis in a matrix. As to the cross-border cooperation, we believe it is a valuable issue for the two Countries.

Linguistic remarks

We double checked with a language expert.

Title

We would keep the original title as our focus is on possible cross-border development cooperation between the two countries. 

Round 2

Reviewer 1 Report

I note that the authors worked hard.

But a version with MS Word corrector mode and so many colors and versions is hard to understand and a bit confusing (it is not the standard in our community - no worries, you can do it).

If I remember well, I gave some literature advices. It seems my offers were not really much pursued, studied and used by the authors. See Erik Kropat, Ayse Ozmen, etc.

Put some nice mathematics it (it is standard today). Do not be afraid of it.

So I am still in favor. But I ask the authors to revisit my last review properly and do all issues carefully and in a positive way of yet not done at all or only done in part.

Please see the recommendations and gain from them.

Please use a reviewer friendly way to show what you have done.

If being the case, I will remain and be positive and acceptive.

Author Response

Dear Reviewer,

As authors we do appreciate your words regarding the hard work that we made to improve the paper and thank you for your valuable suggestions.

We followed your literature advices and put additional references (see number 42), as you suggested. As to mathematics, we do not feel like putting some as it is not our field of expertise and we hope this suits your expectations and does not affect your positive evaluation.

We put all the new version in reviewer friendly way to show what we have done, as you required, and look forward to your kind and precious feedback.

Best regards.

Round 3

Reviewer 1 Report

The paper can be accepted

(If the authors could kind enough to enter some references with authors I named and asked you several times (please see my comments; E. Kropat, A. Ozmen), would be appreciated. 

If you have a chance, please do it now as I asked for a bit more prosperity in modern math. 

Author Response

Dear Reviewer,

We do thank you for all your valuable suggestions to improve the paper. In the previous revisions we added reference 42, as you indicated. In this occasion, we followed your further instructions and added three more references (50, 51 and 52), and hope this suits your indications.

As to your request to more prosperity in modern math, we kindly ask you to consider that we do not really feel confident in doing so, as it is really not our field of expertise. We trust you will understand that we take seriously your indications and we do not feel like invading a field that does not belong to us.

Looking forward to your precious reply, we send our best and warmest wishes.